

# Luteolin supplementation during porcine oocyte maturation improves the developmental competence of parthenogenetic activation and cloned embryos

Pil-Soo Jeong[1], Hae-Jun Yang[1], Se-Been Jeon[1,2], Min-Ah Gwon[1,3], Min Ju Kim[1,2], Hyo-Gu Kang[1,4], Sanghoon Lee[5], Young-Ho Park[1], Bong-Seok Song[1], Sun-Uk Kim[1,6], Deog-Bon Koo[3] and Bo-Woong Sim[1]

[1] Futuristic Animal Resource & Research Center, Korea Research Institute of Bioscience and Biotechnology, Cheongju, Republic of Korea
[2] Department of Animal Science, College of Natural Resources & Life Science, Pusan National University, Miryang, Republic of Korea
[3] Department of Biotechnology, College of Engineering, Daegu University, Gyeongsan, Republic of Korea
[4] Department of Animal Science and Biotechnology, College of Agriculture and Life Science, Chungnam National University, Daejeon, Republic of Korea
[5] Laboratory of Theriogenology, College of Veterinary Medicine, Chungnam National University, Daejeon, Republic of Korea
[6] Department of Functional Genomics, University of Science and Technology, Daejeon, Republic of Korea

Corresponding author
Bo-Woong Sim,
embryont@kribb.re.kr

## ABSTRACT

Luteolin (Lut), a polyphenolic compound that belongs to the flavone subclass of flavonoids, possesses anti-inflammatory, cytoprotective, and antioxidant activities. However, little is known regarding its role in mammalian oocyte maturation. This study examined the effect of Lut supplementation during *in vitro* maturation (IVM) on oocyte maturation and subsequent developmental competence after somatic cell nuclear transfer (SCNT) in pigs. Lut supplementation significantly increased the proportions of complete cumulus cell expansion and metaphase II (MII) oocytes, compared with control oocytes. After parthenogenetic activation or SCNT, the developmental competence of Lut-supplemented MII oocytes was significantly enhanced, as indicated by higher rates of cleavage, blastocyst formation, expanded or hatching blastocysts, and cell survival, as well as increased cell numbers. Lut-supplemented MII oocytes exhibited significantly lower levels of reactive oxygen species and higher levels of glutathione than control MII oocytes. Lut supplementation also activated lipid metabolism, assessed according to the levels of lipid droplets, fatty acids, and ATP. The active mitochondria content and mitochondrial membrane potential were significantly increased, whereas cytochrome c and cleaved caspase-3 levels were significantly decreased, by Lut supplementation. These results suggest that Lut supplementation during IVM improves porcine oocyte maturation through the reduction of oxidative stress and mitochondria-mediated apoptosis.

## INTRODUCTION

Because of their physiological and anatomical similarities to humans, transgenic pigs generated by somatic cell nuclear transfer (SCNT) have been used in many areas of biomedical research, including bioreactors, human disease models, and xenotransplantation (*Gil et al., 2010*; *Meurens et al., 2012*). In parallel, numerous attempts have been made to increase the developmental competence of SCNT embryos, resulting in the increased production of cloned piglets; however, efficiency remains low compared with *in vivo* embryo production (*Grupen, 2014*; *Whitworth & Prather, 2010*). One reason for this low efficiency is the inadequate *in vitro* maturation (IVM) environment of oocytes (*Herrick, 2019*). Oocyte maturation is a complex process that involves both nuclear and cytoplasmic maturation. The low developmental competence of IVM oocytes leads to failed embryonic development, mainly because of improper cytoplasmic maturation despite complete nuclear maturation (*Gilchrist & Thompson, 2007*). Successful cytoplasmic maturation depends on the IVM environment and thus on the medium composition and culture conditions (*Maedomari et al., 2007*; *Marchal et al., 2001*). Optimal IVM conditions are therefore the most important determinant of successful oocyte maturation and a prerequisite for the increased efficiency of transgenic pig production.

Reactive oxygen species (ROS) are commonly generated from byproducts of intracellular energy metabolism, and specifically from the byproducts of mitochondrial respiration in aerobic organisms. Although ROS at moderate levels are essential intracellular signaling molecules involved in normal biological processes, their excessive accumulation (from disrupted redox homeostasis) leads to oxidative stress and thus potentially to cytoskeletal abnormalities, lipid peroxidation, adenosine 5′-triphosphate (ATP) depletion, mitochondrial dysfunction, and apoptosis (*Park et al., 2021*; *Wang et al., 2021*). In the *in vivo* environment, the female reproductive tract (ovary, oviduct, and uterus) protects against ROS accumulation through its antioxidant defense system (*Agarwal et al., 2012*). However, during IVM, an imbalance between ROS generation and antioxidant activity can induce oxidative stress (*Tarin et al., 1996*). Compared with other species, the oocytes of pigs are much more sensitive to physiological stress, such as the stress caused by excess light, abnormal oxygen tension, and sudden changes in pH, because of their high cellular lipid and fatty acid contents (*Sturmey et al., 2009*). Stress-induced excessive ROS generation leads to embryonic arrest or lower embryo viability (*Byrne et al., 1999*). Attempts to reduce oxidative damage by dietary supplementation with numerous ROS scavengers, such as resveratrol, melatonin, and ascorbic acid, have failed to yield IVM conditions compare to the *in vivo* environment (*Kere et al., 2013*; *Lee et al., 2018*). An understanding of oxidative stress defense mechanisms and the application of more effective antioxidants would protect oocytes against oxidative stress while improving IVM.

Flavonoids are polyphenol secondary metabolites widely distributed in fruits, vegetables, and herbs (*de Rijke et al., 2006*). Many animals, including humans, consume

significant amounts of flavonoids, which are present in a wide variety of edible plants (*Mutha, Tatiya & Surana, 2021*). The flavonoid luteolin (3, 4, 5, 7-tetrahydroxyflavone; Lut) belongs to the flavone group and is abundant in many vegetables and fruits, especially broccoli, carrots, and rosemary. Lut has been reported to have many biological functions including anti-inflammatory, antibacterial, anti-cancer, antioxidant, and cytoprotective activities (*Gupta et al., 2018*; *Huang, Kim & Cho, 2023*). Most previous studies of Lut have focused on therapeutic agents due to its potent anti-cancer and anti-inflammatory activities. *Deqiu et al. (2011)* reported that Lut can ameliorate insulin resistance-related endothelial dysfunction. *Chen et al. (2014)* reported that Lut effectively protects mice from lipopolysaccharide-induced lethality by suppressing heat shock protein 90 activity. *Kim et al. (2018)* reported that Lut suppresses airway inflammation by regulating immune response. In addition, the antioxidant activity of Lut reflects its ability to reduce free radical formation and thus oxidative stress in human umbilical vein endothelial cells and neuroblastoma cells (*Chen et al., 2020*; *Zhang et al., 2013*). Although the physiological and biological roles of Lut are well-documented in many cell types, the effect of Lut during meiotic maturation in pigs has not been investigated.

This study examined the effects of Lut supplementation on oocyte maturation and the subsequent developmental competence of SCNT embryos in pigs. Lut was added to the maturation medium; its effects on nuclear maturation, cumulus cell expansion, and subsequent embryonic development after parthenogenetic activation (PA) and SCNT were determined based on intracellular ROS and glutathione (GSH) levels, lipid metabolism, mitochondrial function, and activation of the apoptosis pathway in Lut-treated porcine oocytes.

## MATERIALS AND METHODS

### Ethics statement
This study was conducted in strict accordance with the recommendations of the Korea Research Institute of Bioscience and Biotechnology (KRIBB) Institutional Animal Care and Use Committee (approval no. KRIBB-AEC-21107).

### Chemicals
All chemicals and reagents used in the present study were acquired from Sigma-Aldrich Chemical Company (St. Louis, MO, USA), unless otherwise indicated.

### Oocyte collection and *in vitro* maturation
Oocyte collection and *in vitro* maturation were performed as described previously (*Park et al., 2021*). Porcine ovaries were retrieved from a local slaughterhouse and transported to the laboratory in 0.9% NaCl supplemented with potassium penicillin (75 μg/mL) and streptomycin sulfate (50 μg/mL) at 38.5 °C within 2 h. Cumulus-oocyte complexes (COCs) were recovered from antral follicles (about 3–6 mm in diameter) using an 18-gauge needle fixed to a 10-mL disposable syringe. The COCs were washed three times in 0.9% NaCl supplemented with 0.1% bovine serum albumin (BSA), after which ~50 COCs were matured in 500 μL IVM medium in a four-well multi-dish (Nunc, Roskilde, Denmark) for

44 h at 38.5 °C in 5% $CO_2$ in air. After 22 h, the COCs were cultured in IVM medium without hormones for another 22 h.

### Luteolin (Lut) treatment

The concentration of Lut was chosen according to a previous study (*Park et al., 2021*). Lut was dissolved in dimethyl sulfoxide, then diluted in IVM medium to a final concentration of 5 µM Lut (along with 0.1% dimethyl sulfoxide).

### Assessment of cumulus cell expansion and the nuclear maturation of oocytes

Cumulus cell expansion and nuclear maturation were assessed as described previously, with modifications (*Jeong et al., 2020a*). After 44 h of IVM, the degree of cumulus cell expansion in porcine COCs was observed by morphological examination under a microscope (Nikon Corp., Tokyo, Japan) and classified as follows: grade 1: no expansion; grade 2: poor expansion, with spherical and compacted cumulus cells around the oocyte; grade 3: partial expansion, with the expansion of cumulus cell layers except the corona radiata; grade 4: full expansion of all cumulus cell layers. For assessment of nuclear maturation, cumulus cells after 44 h of IVM were removed by gentle pipetting with 0.1% hyaluronidase in Dulbecco's phosphate-buffered saline (DPBS; Gibco, Grand Island, NY, USA) supplemented with 4 mg/mL BSA. The denuded oocytes were examined under a microscope (Nikon Corp., Tokyo, Japan) and classified as degenerate, immature (without first polar body extrusion), or MII stage.

### Parthenogenetic activation

Parthenogenetic activation was performed as described previously (*Jeong et al., 2017*). MII oocytes in a 1-mm gab wire chamber (CUY5000P1; Nepagene, Chiba, Japan) were parthenogenetically activated with 10 µL of 280 mM mannitol containing 0.1 mM $MgSO_4 \cdot 7H_2O$, 0.1 mM $CaCl_2 \cdot 2H_2O$, 0.5 mM HEPES, and 1 mg/mL polyvinyl alcohol (PVA). The oocytes were activated with an electrical pulse (1.1 kV/cm for 50 µs) that was applied using an electro cell fusion generator (LF101; Nepagene, Chiba, Japan).
The activated oocytes were then transferred into activation medium, which consisted of *in vitro* culture (IVC) medium (porcine zygote medium-3 containing 4 mg/mL BSA) supplemented with 5 µg/mL cytochalasin B and 2 mM 6-dimethylaminopurine, at 38.5 °C under 5% $CO_2$. After 4 h, the oocytes were transferred to IVC medium at 38.5 °C under 5% $CO_2$. Cleavage and blastocyst formation were evaluated at 48 and 144 h, respectively.

### Somatic cell nuclear transfer

SCNT was performed as described previously, with modifications (*Jeong et al., 2020b*, *2021*, *2017*). MII oocytes in DPBS supplemented with 4 mg/mL BSA, 75 µg/mL penicillin G, 50 µg/mL streptomycin sulfate, and 7.5 µg/mL cytochalasin B were cut using a sharp pipette and then the first polar body and cytoplasm-containing chromosomes were removed using the squeezing method under an inverted microscope (DMI 3000B; Leica Microsystems, Wetzlar, Germany) equipped with a micromanipulator (NT-88-V3; Nikon Narishige, Tokyo, Japan). We used KRIBB small miniature pig kidney cells established in a

previous study as donor cells (*Song et al., 2018*). Donor cell injection and fusion were conducted using the Sendai virus (SV)-mediated method. Briefly, the freeze-dried inactivated SV envelope was dissolved in 260 mL of suspending buffer and diluted 1:4 (v/v) with cell fusion buffer, in accordance with the manufacturer's instructions (Cosmo Bio, Osaka, Japan). Donor cells were immersed in the inactivated SV solution for 1 min, then introduced into the perivitelline space of cytoplasts. After 2 h, oocyte-cell couplets that had completely fused, as confirmed using an inverted microscope, were selected and activated for 4 h in activation medium containing 1 μM oxamflatin at 38.5 °C in 5% $CO_2$ in air. The activated embryos were transferred to IVC medium supplemented with 1 μM oxamflatin, cultured for 20 h, washed in fresh IVC medium and then cultured in IVC medium at 38.5 °C in 5% $CO_2$ in air. Cleavage and blastocyst rates were determined at 48 and 144 h, respectively.

## Indirect immunofluorescence assay

Indirect immunofluorescence assay was performed as described previously (*Jeong et al., 2021*). Oocytes or blastocysts were fixed in 4% paraformaldehyde overnight at 4 °C and washed three times for 10 min each in DPBS supplemented with 1 mg/mL PVA (DPBS-PVA). The fixed oocytes/blastocysts were permeabilized by incubation for 1 h at RT in PBS containing 0.5% Triton X-100. After three washes in DPBS-PVA, the oocytes/blastocysts were stored overnight at 4 °C in DPBS-PVA supplemented with 1 mg/mL BSA (DPBS-PVA-BSA). For CDX2 staining, blastocysts were blocked with 10% normal goat serum for an additional 45 min, then incubated at 4 °C overnight with cytochrome c (Abcam, Cambridge, UK), cleaved caspase 3 (Cell Signaling Technology, Danvers, MA, USA) or mouse monoclonal anti-CDX2 (Biogenex Laboratories Inc., San Ramon, CA, USA) as the primary antibody. After three washes in DPBS-PVA-BSA, the oocytes/blastocysts were incubated for 1 h at room temperature (RT) with conjugated secondary antibody, either Alexa-Fluor-488-labeled goat anti-mouse or anti-rabbit IgG (1:200 in DPBS-PVA-BSA), washed three times for 10 min each in DPBS-PVA-BSA, stained for DNA using 2 μg/mL DAPI, and observed using a fluorescence microscope (Olympus, Tokyo, Japan).

## Terminal deoxynucleotidyl transferase-mediated dUTP-digoxygenin nick end-labeling (TUNEL) assay

TUNEL assay was conducted using an *in situ* cell death detection kit (Roche, Basel, Switzerland), as described previously (*Jeong et al., 2021*). Fixed Blastocysts were washed three times for 10 min each in DPBS-PVA and were incubated in DPBS containing 0.5% Triton X-100 for 1h at RT. After three washes for 10 min each in DPBS-PVA, they were incubated in DPBS containing 10 mg/mL BSA for 1 h at RT. Blastocysts were incubated with TUNEL reaction medium for 1 h at 38.5 °C, washed three for 10 min each in DPBS-PVA, and mounted on clean glass slides with DAPI. Whole-mount blastocysts were observed under a fluorescence microscope (Olympus, Tokyo, Japan) by using detection for DAPI labeled (total cell numbers) or TUNEL positive (apoptotic cell numbers) nuclei. Five to 10 blastocysts per treatment group were used in the TUNEL assay in each independent experiment.

## Measurement of intracellular ROS and GSH levels

The oocytes were washed three times with DPBS-PVA and incubated in DPBS-PVA containing 5 µM CM-H2DCFDA (Invtrogene, Carlsbad, CA, USA) for 30 min or 10 µM CMF2HC (Invitrogen, Waltham, MA, USA) for 10 min. After incubation, the oocytes were washed three times with DPBS-PVA. Fluorescence images were captured with a fluorescence microscope (DMi8; Leica Microsystems) equipped with an ultraviolet filter (460 nm for ROS or 370 nm for GSH). Fluorescence intensity was analyzed using ImageJ software (version 1.47; National Institutes of Health, Bethesda, MD, USA).

## Analysis of ATP, lipid droplet, and fatty acid contents

ATP, lipid droplet, and fatty acid analysis were performed as described previously (*Joo et al., 2023*). The oocytes were washed three times in DPBS-PVA and fixed in 4% paraformaldehyde overnight at 4 °C. For ATP staining, the fixed oocytes were washed three times in DPBS-PVA and incubated for 1 h at RT in the dark in DPBS containing 500 nM BODIPY FL ATP (BODIPY-ATP; A12410; Molecular Probes, Eugene, OR, USA). Subsequently, the oocytes were washed three times with DPBS-PVA, mounted on clean glass slides, and stained with DAPI. For lipid droplet and fatty acid staining, the fixed oocytes were washed three times in DPBS-PVA and incubated for 1 h at RT in the dark in DPBS containing 10 µg/mL BODIPY 493/503 (BODIPY-LD; D3922; Molecular Probes, Eugene, OR, USA) or 6 µM BODIPY 558/568 C12 (BODIPY-FA; D3835; Molecular Probes, Eugene, OR, USA). After three washes in DPBS-PVA, the oocytes were mounted on glass slides and imaged using a fluorescence microscope (DMi8; Leica Microsystems, Wetzlar, Germany). Fluorescence intensity was analyzed using ImageJ software after data normalization *via* subtraction of background intensity from each oocyte size.

## Analysis of mitochondria content and mitochondrial membrane potential

MitoTracker and JC-1 staining were performed as described previously (*Jeong et al., 2019*). The oocytes were incubated for 1 h in DPBS-PVA containing MitoTracker Red CMXRos (Invitrogen, Waltham, MA, USA) or JC-1 (1:100) (Cayman Chemical, Ann Arbor, MI, USA), washed three times for 10 min each in DPBS-PVA, and fixed for 2 h at RT in 4% paraformaldehyde. After three more 10 min washes in DPBS-PVA, the oocytes were either immediately observed under a fluorescence microscope (DMi8; Leica Microsystems, Wetzlar, Germany) for JC-1 staining or mounted on glass slides in Vectashield containing DAPI (Vector Laboratories, Newark, CA, USA) and then observed under a fluorescence microscope (DMi8; Leica Microsystems, Wetzlar, Germany) for MitoTracker staining. Fluorescence intensity was analyzed using ImageJ software after data normalization *via* subtraction of background intensity from each oocyte size.

## Statistical analyses

All experiments were repeated at least three times. The results are expressed as means ± standard errors of the mean. Data were analyzed by analysis of variance, followed by

Student's t-test, using SigmaStat software (SPSS Inc., Chicago, IL, USA). *P*-values < 0.05 were considered indicative of statistical significance.

## RESULTS

### Luteolin enhances oocyte maturation and cumulus cell expansion in porcine COCs

To investigate the effects of Lut on meiotic progression in porcine oocytes, COCs were cultured in IVM medium with or without 5 μM Lut during IVM; cumulus cell expansion and oocyte nuclear maturation were examined after 44 h. A comparison of the control and Lut-supplemented groups revealed no significant differences in the proportions of grade 1 (Con, 2.0 ± 0.7% *vs.* Lut, 1.0 ± 0.4%) and grade 2 (Con, 3.3 ± 0.7% *vs.* Lut, 2.7 ± 0.8%) cumulus cell expansion (Figs. 1A, 1C and 1D). However, Lut supplementation significantly decreased the proportion of cumulus cells with partial expansion (grade 3; Con, 27.3 ± 2.6% *vs.* Lut, 20.3 ± 1.7%) and increased the proportion of cumulus cells with full expansion (grade 4; Con, 67.3 ± 2.5% *vs.* Lut, 76.0 ± 2.0%). Lut supplementation also significantly increased the proportion of MII oocytes (Con, 81.7 ± 1.1% *vs.* Lut, 89.3 ± 0.8%) and decreased the proportion of immature oocytes (Con, 11.3 ± 1.0% *vs.* Lut, 5.3 ± 0.4%); compared with the control group, there was no significant difference in the proportion of degenerated oocytes (Con, 7.0 ± 1.1% *vs.* Lut, 5.3 ± 0.7%) (Figs. 1B, 1C and 1E). These results suggest that Lut supplementation during IVM improves the meiotic maturation of porcine oocytes.

### Luteolin supplementation during IVM improves the developmental competence of porcine PA embryos

PA embryos derived from Lut-supplemented MII oocytes exhibited significantly higher rates of cleavage (Con, 79.3 ± 2.4% *vs.* Lut, 88.3 ± 1.0%) and blastocyst formation (Con, 36.8 ± 3.1% *vs.* Lut, 50.2 ± 2.0%), compared with control embryos (Figs. 2A–2C). Given the beneficial effects of Lut supplementation during IVM on PA embryonic development, post-blastulation development, inner cell mass (ICM)/trophectoderm (TE) cell number, and apoptosis rate were also evaluated as indicators of blastocyst quality. Within the Lut-supplemented group, the proportion of middle stage blastocysts was considerably lower (Con, 33.4 ± 2.4% *vs.* Lut, 17.9 ± 1.5%), whereas the proportion of expanded stage blastocysts was higher (Con, 50.2 ± 2.0% *vs.* Lut, 74.9 ± 3.2%), than in the control group (Figs. 2D and 2E). CDX2 staining revealed no difference in ICM cell number (Con, 10.0 ± 0.7 *vs.* Lut, 10.3 ± 0.7); in the Lut supplementation group, both TE (Con, 23.2 ± 1.6 *vs.* Lut, 28.7 ± 1.3) and the total cell number (Con, 33.2 ± 2.0 *vs.* Lut, 39.0 ± 1.4) were significantly increased compared with the control group (Figs. 2F and 2G). The TUNEL assay results showed that Lut supplementation significantly reduced the apoptotic cell number (Con, 1.9 ± 0.2 *vs.* Lut, 1.3 ± 0.2) and the apoptosis rate (Con, 5.6 ± 0.7% *vs.* Lut, 3.2 ± 0.4%) (Figs. 2H–2J).

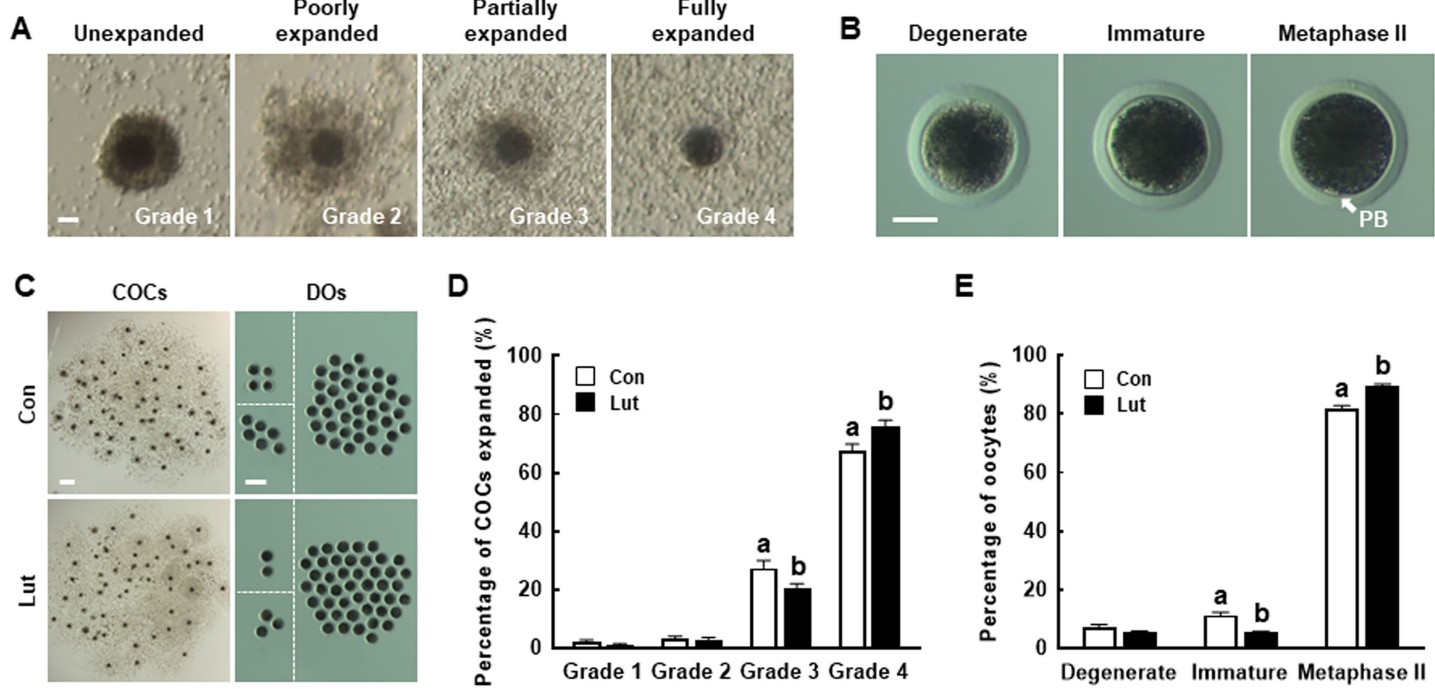

**Figure 1 Effects of luteolin (Lut) supplementation on porcine oocyte maturation.** Morphological classes of (A) cumulus cell expansion and (B) denuded oocytes (DOs) in cumulus-oocyte complexes (COCs). Bar = 50 μm. (C) Representative images of COCs (left, bar = 400 μm) and DOs (right, bar = 200 μm) in the control and Lut-treated groups after 44 h of *in vitro* maturation (IVM). (D) The percentage of cumulus cell expansion and (E) the proportions of different stages of nuclear maturation in the control and Lut-treated groups ($n$ = 300 per group). Data are from six independent experiments and different superscript letters indicate significant differences ($P < 0.05$).

## Luteolin supplementation during IVM improves the developmental competence of porcine SCNT embryos

The effect of Lut supplementation during IVM on developmental competence after SCNT was also investigated. Consistent with the result for PA, SCNT embryos from Lut-supplemented MII oocytes exhibited significantly increased rates of cleavage (Con, 79.6 ± 1.7% *vs.* Lut, 85.2 ± 1.4%) and blastocyst formation (Con, 36.2 ± 2.2% *vs.* Lut, 45.2 ± 2.4%), compared with the controls (Figs. 3A–3C). Although the proportions of early (Con, 18.7 ± 3.7% *vs.* Lut, 10.9 ± 4.5%) and middle (Con, 33.7 ± 5.8% *vs.* Lut, 25.3 ± 4.5%) blastocyst stages did not significantly differ between the control and Lut-supplemented groups, a large increase in the proportion of the hatched (hatching or already hatched; Con, 47.6 ± 3.4% *vs.* Lut, 63.8 ± 1.1%) blastocyst stage was observed in the Lut-supplemented group (Figs. 3D and 3E). According to the CDX2 staining results, there was no difference in ICM cell number (Con, 4.8 ± 0.4 *vs.* Lut, 5.2 ± 0.3), but both TE (Con, 24.6 ± 1.0 *vs.* Lut, 28.4 ± 0.9) and total cell number (Con, 29.4 ± 1.2 *vs.* Lut, 33.5 ± 0.9) were significantly increased in the Lut supplementation group compared with the control group (Figs. 3F and 3G). In the TUNEL assay, the number of apoptotic cells (Con, 2.2 ± 0.2 *vs.* Lut, 1.2 ± 0.2) and the apoptosis rate (Con, 6.6 ± 0.7% *vs.* Lut, 3.8 ± 0.6%) were significantly reduced in the Lut supplementation group (Figs. 3H–3J). These results provided further

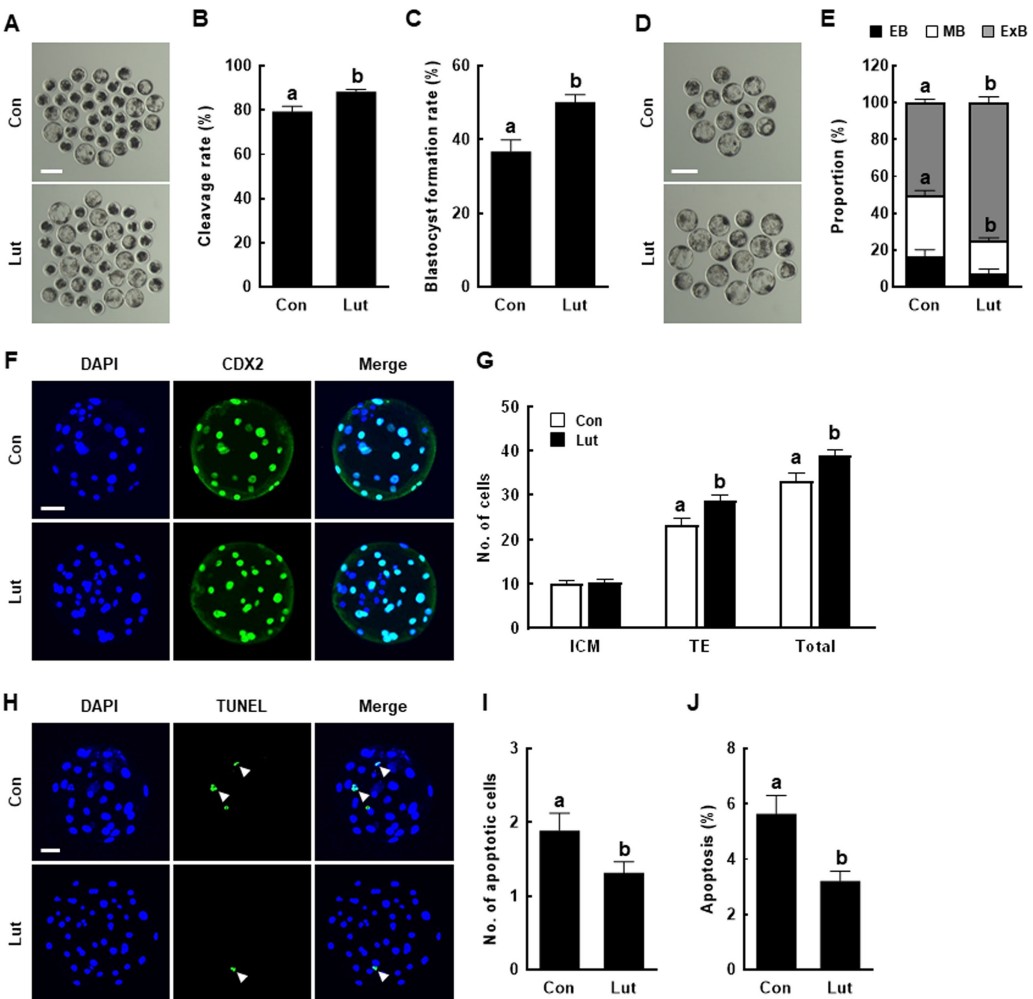

**Figure 2 Effects of Lut supplementation during IVM on the developmental competence after parthenogenetic activation (PA).** (A) Representative images of day 6 PA embryos from control and Lut-treated oocytes. Bar = 200 μm. (B) Cleavage rates and (C) blastocyst formation rates of embryos from control and Lut-treated oocytes after PA (Con; $n = 208$, Lut; $n = 221$). (D) Representative images of blastocysts and (E) quantification of the proportion of each blastocyst stage from control and Lut-treated oocytes after PA (Con; $n = 77$, Lut; $n = 111$). Bar = 200 μm. (F) Representative images of CDX2 staining. Bar = 50 μm. (G) Quantification of the inner cell mass (ICM) and trophectoderm (TE) as well as total cell numbers ($n = 30$ per group). (H) Representative images of the TUNEL assay results. Green and blue fluorescence indicate TUNEL-labeled embryos (white arrow) and nuclei, respectively. Bar = 50 μm. Quantification of the (I) apoptotic cell number and (J) apoptosis rate ($n = 35$ per group). Data are from five independent experiments and different superscript letters indicate significant differences ($P < 0.05$).

evidence of the beneficial effects of Lut supplementation during IVM on porcine embryonic development after PA and SCNT.

## Luteolin prevents intracellular ROS accumulation in porcine oocytes

Analysis of the antioxidant properties of Lut during IVM revealed lower ROS levels (Con, 78.2 ± 4.4 *vs.* Lut, 50.7 ± 3.0) and higher GSH levels (Con, 66.4 ± 1.5 *vs.* Lut, 78.6 ± 1.5) in Lut-supplemented oocytes than in control MII oocytes (Figs. 4A and 4B).

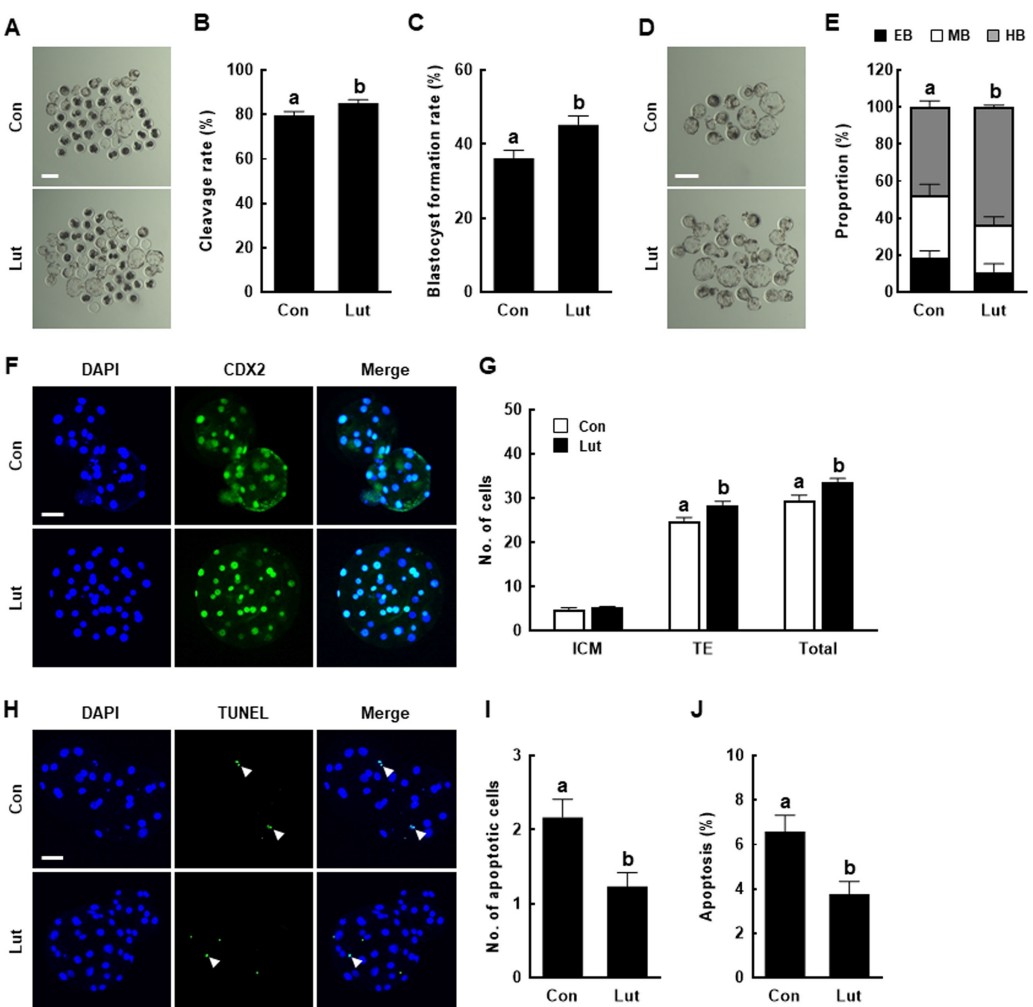

**Figure 3 Effects of Lut supplementation during IVM on developmental competence after somatic cell nuclear transfer (SCNT).** (A) Representative images of day 6 SCNT embryos from control and Lut-treated oocytes. Bar = 200 μm. (B) The cleavage rate and (C) blastocyst formation rates of embryos from control and Lut-treated oocytes after SCNT (Con; $n = 183$, Lut; $n = 190$). (D) Representative images of blastocysts and (E) quantification of the proportion of each blastocyst stage from control and Lut-treated oocytes after SCNT (Con; $n = 67$, Lut; $n = 86$). Bar = 200 μm. (F) Representative images of CDX2 staining. Bar = 50 μm. (G) Quantification of ICM and TE, as well as total cell numbers ($n = 34$ per group). (H) Representative images of the TUNEL assay results. Green and blue fluorescence indicate TUNEL-labeled embryos (white arrow) and nuclei, respectively. Bar = 50 μm. Quantification of (I) apoptotic cell number and (J) the apoptosis rate ($n = 30$ per group). Data are from five independent experiments and different superscript letters indicate significant differences ($P < 0.05$).

## Luteolin enhances lipid metabolism of porcine oocytes

Because lipid metabolism contributes to the energy needed during oocyte maturation and embryonic development, the lipid droplet, fatty acid, and ATP contents of porcine oocytes supplemented with Lut during IVM were measured. Lut-supplemented MII oocytes showed significantly higher the intensity of lipid droplets (Con, 67.3 ± 2.1 *vs.* Lut, 80.0 ± 1.8) and fatty acids (Con, 54.1 ± 3.8 *vs.* Lut, 70.9 ± 4.8) compared to control (Figs. 4C and

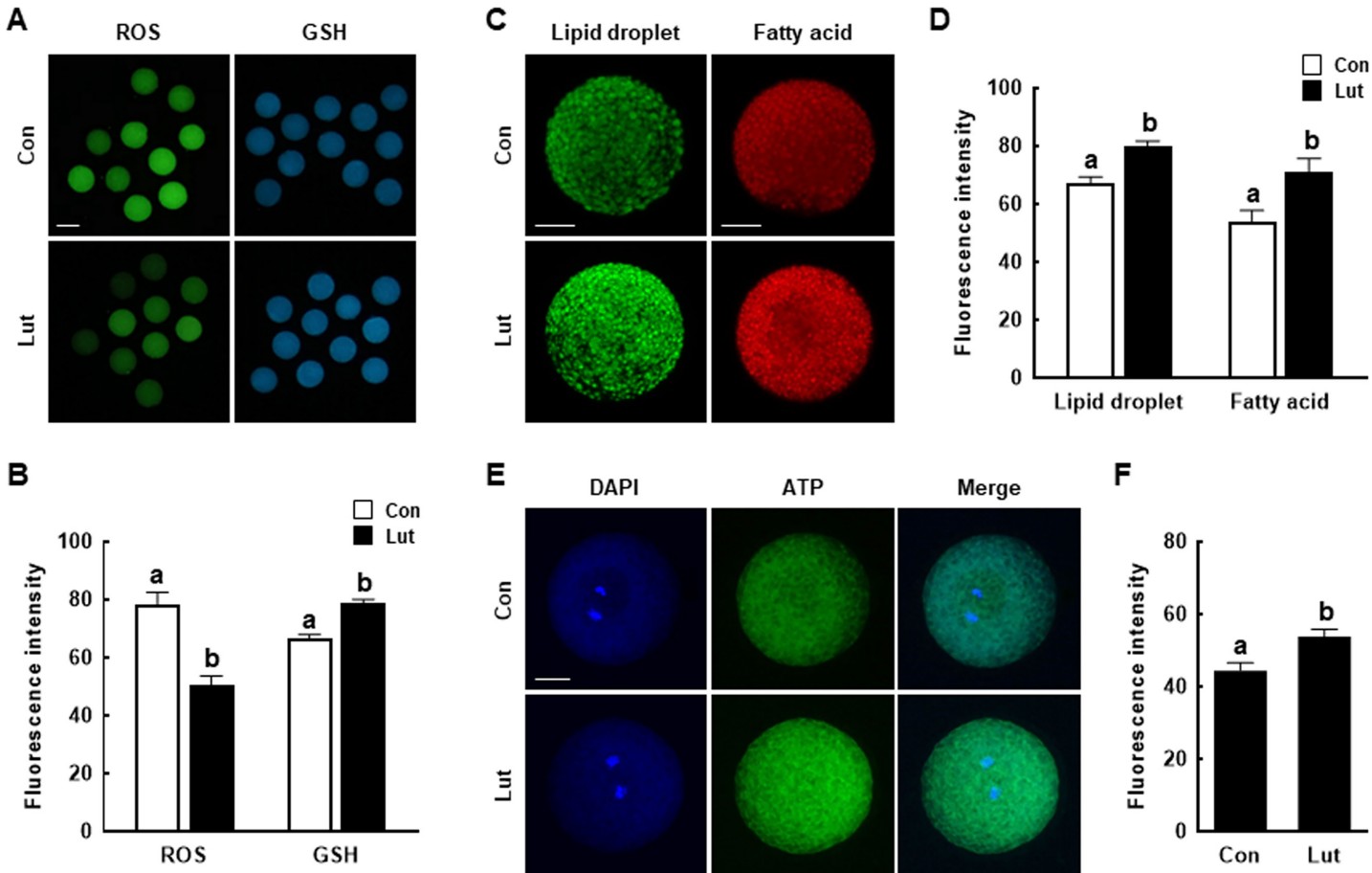

**Figure 4 Effects of Lut supplementation during IVM on oxidative stress and lipid metabolism.** (A) Fluorescence images of oocytes treated with CM-H2DCFDA to measure the intracellular levels of reactive oxygen species (left) and with CMF2HC to measure the intracellular levels of glutathione (right) in the indicated groups. Bar = 100 um. (B) Quantification of fluorescence intensity ($n = 50$ per group). (C) Fluorescence images of oocytes stained with BODIPY-LD (left) and BODIPY-FA (right). Bar = 50 um. (D) Quantification of fluorescence intensity (LD: lipid droplets; $n = 30$ per group, FA, fatty acids; $n = 25$ per group). (E) Fluorescence images of BODIPY-ATP-stained oocytes. Bar = 50 um. (F) Quantification of fluorescence intensity ($n = 35$ per group). Data are from three independent experiments and different superscript letters indicate significant differences ($P < 0.05$).

4D). Moreover, ATP content was higher in Lut-supplemented MII oocytes than control (Con, 44.5 ± 2.1 *vs.* Lut, 53.8 ± 2.0) (Figs. 4E and 4F).

## Luteolin enhances the mitochondrial functions of porcine oocytes

As sites of oxidative phosphorylation, mitochondria are essential to cellular energy production during oocyte maturation and subsequent embryonic development. Accordingly, the effects of Lut on the developmental competence of porcine oocytes during IVM were examined by measuring both the mitochondrial content and mitochondrial membrane potential. Compared with the control group, the mitochondrial content of Lut-supplemented oocytes was higher (Con, 53.8 ± 2.8 *vs.* Lut, 66.6 ± 3.9) (Figs. 5A and 5B) and the J-aggregate (high membrane potential)/J-monomer (low membrane potential) ratio was significantly increased (Con, 1.50 ± 0.01 *vs.* Lut, 1.64 ± 0.01) (Figs. 5C and 5D).

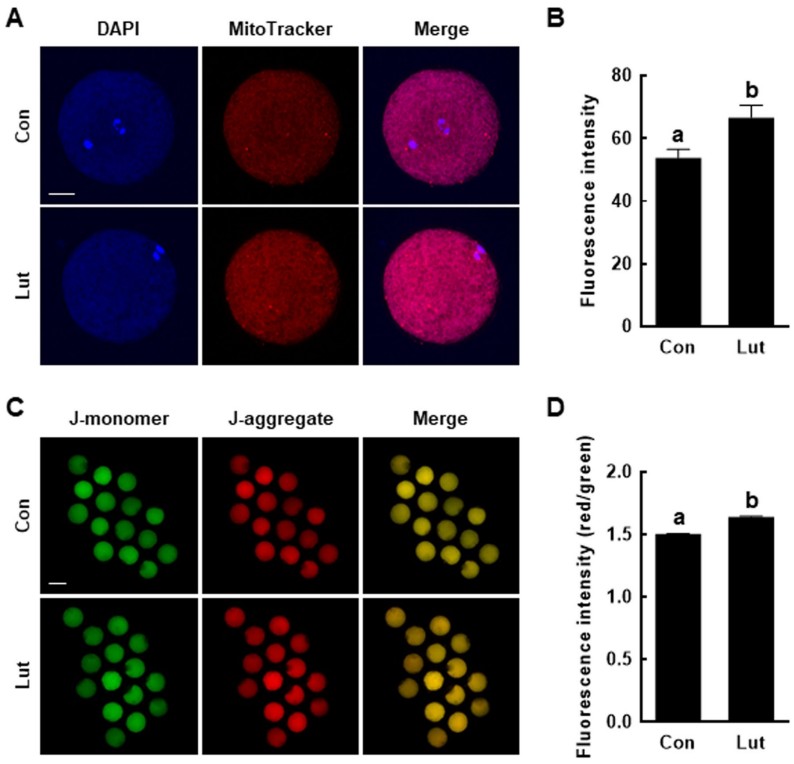

**Figure 5 Effects of Lut supplementation during IVM on mitochondrial function.** (A) Fluorescence images of oocytes stained with MitoTracker. Bar = 100 um. (B) Quantification of fluorescence intensity ($n$ = 44 per group). (C) Fluorescence images of oocytes stained with JC-1. Bar = 100 um. (D) Quantification of the (red/green) fluorescence intensity ($n$ = 45 per group). Data are from three independent experiments and different superscript letters indicate significant differences ($P < 0.05$).

## Luteolin protects mitochondrial-mediated apoptosis of porcine oocytes

The effects of Lut on the apoptosis pathway were assessed by measuring the levels of cytochrome c and cleaved caspase-3 in porcine MII oocytes. Cytochrome c levels were lower in Lut-supplemented oocytes than in control MII oocytes (Con, 85.9 ± 3.1 *vs.* Lut, 76.7 ± 2.1) (Figs. 6A and 6B); cleaved caspase-3 levels were also lower in Lut-supplemented oocytes (Con, 10.3 ± 0.4 *vs.* Lut, 8.8 ± 0.4) (Figs. 6C and 6D).

## DISCUSSION

The production of transgenic pigs by SCNT has attracted interest from biomedical researchers; however, the production efficiency resulting from this approach is suboptimal because of factors related to oocyte quality and culture conditions. Poor IVM conditions are mainly the result of the oxidative stress induced by the overproduction of ROS, which disrupts normal cell structures and processes, including cell membranes, RNA transcription, DNA/protein synthesis, and lipid metabolism, leading to decreased oocyte quality (*Guerin, El Mouatassim & Menezo, 2001*). Previous studies showed that antioxidant supplementation during IVM allows the successful cytoplasmic maturation of

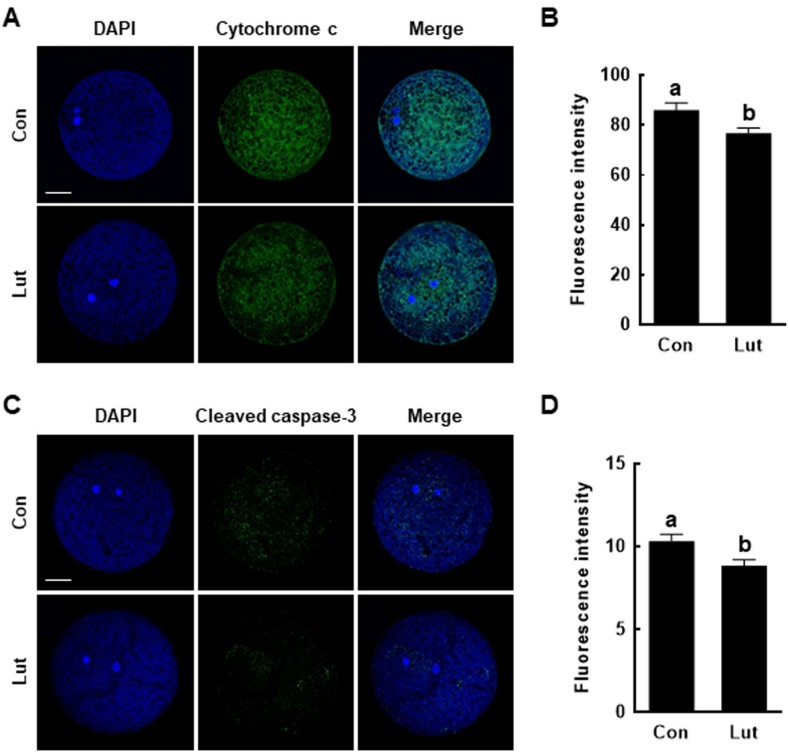

**Figure 6 Effects of Lut supplementation during IVM on mitochondria-mediated apoptosis.** (A) Fluorescence images of oocytes stained with cytochrome c. Bar = 100 um. (B) Quantification of fluorescence intensity ($n = 26$ per group). (C) Fluorescence images of oocytes stained with cleaved caspase-3. Bar = 100 um. (D) Quantification of fluorescence intensity ($n = 28$ per group). Data are from three independent experiments and different superscript letters indicate significant differences ($P < 0.05$).

porcine oocytes and enhances the developmental capacity of SCNT embryos by reducing ROS levels (*Kere et al., 2013*; *Qi et al., 2020*; *Taweechaipaisankul et al., 2016*). Lut is a powerful antioxidant with excellent radical-scavenging and cytoprotective abilities (*Heim, Tagliaferro & Bobilya, 2002*). Luteolin has a double bond between C2 and C3 that provides hydrogen/electrons, and can stabilize the radical group by blocking the Fenton reaction using the oxo group of C4 that binds to transition metal ions such as iron and copper (*Gendrisch et al., 2021*). These structural features of Lut prevent oxidative damage by inhibiting pro-oxidant enzymes and inducing antioxidant enzymes. A previous study reported that Lut upregulates the transcription of ROS scavenging enzymes by activating the SIRT and FOXO families in human monocytes (*Kim, Lee & Yun, 2017*). Moreover, the antioxidant abilities of Lut are synergistically reinforced by its interactions with other antioxidants, including vitamins and cellular redox systems (*Wolfle, Haarhaus & Schempp, 2013*). The results of this study demonstrated that Lut supplementation during IVM improves porcine oocyte maturation and subsequent embryonic development after PA and SCNT. These effects can be at least partly attributed to the benefits conferred by the reduction of oxidative stress on lipid metabolism and mitochondrial function.

The cumulus cells surrounding the oocyte provide nutrients, hormones, and an optimal microenvironment for oocyte growth and maturation (*Assidi, Dieleman & Sirard, 2010*). A previous study showed that the removal of cumulus cells before IVM decreases the oocyte maturation rate and subsequent embryonic development after fertilization, indicating that cumulus cells are crucial for oocyte maturation and quality (*Wongsrikeao et al., 2005*). Cumulus cell expansion has been used as a morphological indicator of oocyte quality, based on the direct correlation of the optimal expansion of cumulus cells with proper oocyte maturation and the acquisition of developmental competence (*Brown et al., 2017*). Cumulus cells also play an important role in protecting oocytes from environmental stress, including oxidative stress (*Tatemoto, Sakurai & Muto, 2000*). However, excessive oxidative stress in cumulus cells induced cellular apoptosis and reduced CX43 levels (gap junction related factors), resulting in reduced oocyte quality (*Qian et al., 2022*). Moreover, oxidative stress caused by hydrogen peroxide exposure reduced not only expression of cumulus cell expansion- and maternal factor-related genes but also blastocyst formation and total cell number; interestingly, isorhamnetin treatment, another flavonoid compound, rescued by these defects (*Oh et al., 2023*). In this study, Lut supplementation prevents oxidative stress in porcine COCs by reducing ROS levels and increasing GSH levels. Moreover, Lut supplementation during IVM significantly increased the rate of COCs with fully expanded cumulus cells and MII oocytes, which then improved porcine IVM efficiency by reducing oxidative stress.

Oocyte quality is gradually acquired during ovarian follicular development, which includes oocyte growth and maturation (*Eppig et al., 1994*). Both preimplantation embryonic development and pregnancy maintenance to full term can be traced to oocyte quality. Although many transgenic pigs obtained by SCNT have been produced from IVM-derived oocytes, only a small percentage of those oocytes develop to the blastocyst stage, suggesting that current IVM systems do not adequately support the acquisition of oocyte developmental competence (*Trounson, Anderiesz & Jones, 2001*). Improving oocyte quality would thus increase the rate of successful embryonic development after SCNT. This study showed that, in PA and SCNT embryos, Lut supplementation during IVM significantly increases the cleavage rate, blastocyst formation, proportions of expanded or hatching blastocysts, cell number, and cellular survival rate; these results demonstrate the beneficial effects of Lut on the developmental competence of PA and SCNT embryos through enhancement of oocyte quality.

Lipid droplets are crucial organelles that store oocyte intracellular lipids to regulate cellular metabolism and energy homeostasis. During oocyte maturation, oocytes synthesize lipid droplets using fatty acids, liquid-liquid phase separation and lipid accumulation in the membrane of the endoplasmic reticulum. Simultaneously, fatty acids are cleaved from triglycerides by lipase activation and transported into the mitochondria, where they undergo β-oxidation for the ATP production that is necessary to support successful oocyte growth and maturation (*Ferguson & Leese, 2006*). Abnormal lipid metabolism, including elevated levels of saturated fatty acids, impairs oocyte maturation and quality, thus interrupting subsequent embryonic development (*Bradley & Swann, 2019*). Immature porcine oocytes are particularly sensitive to the damage caused by
oxidative stress because their lipid contents are much higher than the contents in other mammalian species, including cows, mice, and sheep, suggesting an important role for lipid metabolism in porcine oocyte maturation (*McEvoy et al., 2000*; *Somfai et al., 2011*). The significantly higher contents of lipid droplets, fatty acids, and ATP in Lut-supplemented oocytes, compared with control MII oocytes, suggest that the improved IVM efficiency achieved by Lut supplementation is mediated by enhancement of lipid metabolism. However, further detailed experiments are needed to provide direct evidence of the effects of Lut supplementation on lipid metabolism during porcine oocyte maturation.

Mitochondria are cytoplasmic organelles that regulate ATP synthesis, ROS generation, calcium signaling, fatty acid oxidation, apoptosis, and other cellular metabolic processes (*McBride, Neuspiel & Wasiak, 2006*). They also support important aspects of mammalian reproduction, including oocyte maturation, fertilization, and embryonic development (*Van Blerkom, 2011*). However, oxidative stress by excessive ROS accumulation triggers a decrease in cellular mitochondrial content through the suppression of mitochondrial biogenesis involving mtDNA replication, membrane formation, and mitochondrial division (*Bouchez & Devin, 2019*; *Nisoli et al., 2004*). Oxidative stress also disrupts the mitochondrial membrane potential, which reduces the oxidative phosphorylation reaction and ATP synthesis (*Aiken, Cindrova-Davies & Johnson, 2008*). Numerous studies have reported that antioxidant supplementation during IVM is commonly able to overcome mitochondria injury by reducing ROS levels (*Chen et al., 2022*; *Li et al., 2022*; *Liu et al., 2023*). A previous study reported that Lut increased PGC-1α and nitric oxide levels, which would correct cellular impairment and may reduce mitochondrial ROS production, suggesting that Lut may promote mitochondrial biogenesis (*Suh, Chon & Choi, 2016*). Our results showed that Lut-supplemented oocytes not only reduced ROS levels but also increased mitochondrial number and membrane potential compared to the control, indicating that Lut supplementation increased mitochondrial number and function due to its antioxidant activity. In addition, oxidative stress disrupts the mitochondrial membrane potential, resulting in the release of cytochrome c from the mitochondria into the cytoplasm (*Liu et al., 1996*). Cytochrome c is a member of the mitochondria-dependent apoptotic pathway; its activation triggers the caspase cascade, including caspase-9 and caspase-3, to induce apoptosis (*Sun et al., 2018*). A previous study reported that Lut suppressed lysophosphatidylcholine-induced apoptosis by blocking calcium/mitochondria/caspase-dependent pathway in human umbilical vein endothelial cells (*Song et al., 2010*). In this study, Lut supplementation significantly reduced the levels of cytochrome c and cleaved caspase-3 compared to the control, thereby preventing apoptosis. Taken together, these results indicate that Lut supplementation during IVM improves oocyte quality by enhancing mitochondrial function through the reduction of intracellular ROS levels.

## CONCLUSIONS

Our results demonstrate that Lut supplementation during IVM improves porcine oocyte quality and subsequent embryonic development after PA and SCNT. The positive effects of

Lut can be attributed to the reduction of oxidative stress and enhancements of both lipid metabolism and mitochondrial function. These findings strongly suggest that Lut supplementation can improve the production of high-quality porcine oocytes, thus be used to improve the production of transgenic pigs for biomedical research.

### Funding

This research was supported by the Korea Research Institute of Bioscience and Biotechnology (KRIBB) Research Initiative Program (KGM4252331) and the National Research Foundation of Korea (NRF) grant funded by the Ministry of Science and ICT (MSIT) (2021M3H9A1096895), Republic of Korea. The funders had no role in study design, data collection and analysis, decision to publish, or preparation of the manuscript.

### Grant Disclosures

The following grant information was disclosed by the authors:
Korea Research Institute of Bioscience and Biotechnology (KRIBB) Research Initiative Program: KGM4252331.
National Research Foundation of Korea (NRF).
Ministry of Science and ICT (MSIT): 2021M3H9A1096895.

### Competing Interests

The authors declare that they have no competing interests.

### Author Contributions

- Pil-Soo Jeong conceived and designed the experiments, performed the experiments, analyzed the data, prepared figures and/or tables, authored or reviewed drafts of the article, and approved the final draft.
- Hae-Jun Yang performed the experiments, prepared figures and/or tables, and approved the final draft.
- Se-Been Jeon performed the experiments, prepared figures and/or tables, and approved the final draft.
- Min-Ah Gwon performed the experiments, prepared figures and/or tables, and approved the final draft.
- Min Ju Kim performed the experiments, prepared figures and/or tables, and approved the final draft.
- Hyo-Gu Kang performed the experiments, prepared figures and/or tables, and approved the final draft.
- Sanghoon Lee performed the experiments, authored or reviewed drafts of the article, and approved the final draft.
- Young-Ho Park analyzed the data, authored or reviewed drafts of the article, and approved the final draft.
- Bong-Seok Song analyzed the data, authored or reviewed drafts of the article, and approved the final draft.

- Sun-Uk Kim conceived and designed the experiments, authored or reviewed drafts of the article, and approved the final draft.
- Deog-Bon Koo conceived and designed the experiments, authored or reviewed drafts of the article, and approved the final draft.
- Bo-Woong Sim conceived and designed the experiments, analyzed the data, prepared figures and/or tables, authored or reviewed drafts of the article, and approved the final draft.

## Data Availability

The raw measurements are available in the Supplemental File.

## Supplemental Information

Supplemental information for this article can be found online at http://dx.doi.org/10.7717/peerj.15618#supplemental-information.

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
