# Peer review of "Luteolin supplementation during porcine oocyte maturation improves the developmental competence of parthenogenetic activation and cloned embryos"

_PeerJ, doi:10.7717/peerj.15618_

## Round 0.1 · original submission · Minor Revisions

I apologise for the length of time it has taken to reach a decision on this. Enclosed please see two sets of comments - both of which contain what are I think largely straightforward suggestions to improve your work. Please take cognisance of these and revise your manuscript accordingly. Set out your changes in a cover letter.

I look forward to seeing the revised paper.

·

Basic reporting

Luteolin research background is not sufficiently reviewed, especially current status on its molecular functional role in a variety of signaling pathways, and its implication in oocyte maturation and quality.
Discussion should be thorough (functional role of luteolin, increase of lipid droplets, enhancement of lipid metabolism etc.). Only were conclusive statements drawn, instead of detailed explanation and interpretation.

Huang L, Kim MY, Cho JY. Immunopharmacological Activities of Luteolin in Chronic Diseases. Int J Mol Sci. 2023 Jan 21;24(3):2136. doi: 10.3390/ijms24032136. PMID: 36768462; PMCID: PMC9917216.
Liu X, Meng J. Luteolin alleviates LPS-induced bronchopneumonia injury in vitro and in vivo by down-regulating microRNA-132 expression. Biomed Pharmacother. 2018 Oct;106:1641-1649. doi: 10.1016/j.biopha.2018.07.094. Epub 2018 Jul 30. PMID: 30119240.
Prasher P, Sharma M, Singh SK, Gulati M, Chellappan DK, Zacconi F, De Rubis G, Gupta G, Sharifi-Rad J, Cho WC, Dua K. Luteolin: a flavonoid with a multifaceted anticancer potential. Cancer Cell Int. 2022 Dec 8;22(1):386. doi: 10.1186/s12935-022-02808-3. PMID: 36482329; PMCID: PMC9730645.
Qiao XR, Feng T, Zhang D, Zhi LL, Zhang JT, Liu XF, Pan Y, Xu JW, Cui WJ, Dong L. Luteolin alleviated neutrophilic asthma by inhibiting IL-36γ secretion-mediated MAPK pathways. Pharm Biol. 2023 Dec;61(1):165-176. doi: 10.1080/13880209.2022.2160770. PMID: 36604842; PMCID: PMC9828607.

Experimental design

no comment

Validity of the findings

Authors only stated their major findings related to luteolin treatment on oocyte maturation and quality.
Since literature is not properly reviewed, the rationale, benefit and significance of their study is not clearly stated.

Additional comments

Major points
1. Line 97-98: As stated before, functional role of luteolin should be thoroughly reviewed.
2. Higher intensity of lipid droplets indicates not the enhancement of lipid metabolism. Authors should examine in detail whether the number of lipid droplets increases, but the size decreases, and explore the underlying mechanism.
3. Mitochondrial numbers also increase. How does luteolin relate to this?
3. Discussion only focused on the results, or simple re-statement. Details on the functional role of luteolin and the relationship with the results obtained should be discussed.

Minor points
1. Title too long.
2. Line 94: "Among its reported biological "
3. Cleavage rates in Figure 2B and 3B look similar to each other. Please check!

Reviewer 2 ·

Basic reporting

No comment

Experimental design

No comment

Validity of the findings

No comment

Additional comments

In this manuscript “Luteolin supplementation during porcine oocyte maturation improves the developmental competence of parthenogenetic activation and cloned embryos by regulating lipid metabolism and mitochondrial function”, the authors investigated the effect of luteolin supplementation of in vitro maturation (IVM) on oocyte maturation and subsequent developmental competence after parthenogenetic activation (PA) and somatic cell nuclear transfer (SCNT) in pigs. After PA and SCNT, developmental competence of luteolin-supplemented oocytes, including cleavage, blastocyst formation, and blastocyst cell numbers was significantly increased. ROS levels were significantly decreased and lipid metabolism such as the levels of lipid droplets, fatty acid, and ATP were increased in luteolin-supplemented oocytes. Thereby, mitochondrial membrane potential was significantly enhanced and apoptosis levels reflected by cytochrome c and cleaved caspase-3 levels were significantly reduced. These changes could support the conclusion of this study that luteolin improved porcine oocyte maturation and subsequent embryonic development by regulating lipid metabolism and mitochondrial function. It is an interesting piece of work that provides useful information for developing porcine in vitro oocyte maturation system. However, there are some points to be revised, as listed in specific comments.

Comments
1. Please insert page numbers in the manuscript.
2. In line 124, the concentration of luteolin was chosen according to “a” previous study. Please revise this.
3. The authors evaluated the levels of ROS, cytochrome c, and cleaved caspase-3 in luteolin-supplemented porcine oocytes. To confirm the changes of apoptosis level in oocyte, TUNEL assay could be conducted using oocyte samples.
4. ROS levels were already measured in this study. How about GSH levels? Have the authors ever conducted GSH measurement using luteolin-supplemented oocytes? GSH is one of the indicators for oocyte cytoplasmic maturation. GSH content could be changed by changes in ROS levels through antioxidant supplementation.

---

## Round 0.2 · accepted · Accept

Thank you for your careful response. I am happy to accept the work now.